# Genomics of Particulate Matter Exposure Associated Cardiopulmonary Disease: A Narrative Review

**DOI:** 10.3390/ijerph16224335

**Published:** 2019-11-07

**Authors:** Julia Citron, Emma Willcocks, George Crowley, Sophia Kwon, Anna Nolan

**Affiliations:** 1Department of Medicine, Division of Pulmonary, Critical Care and Sleep Medicine, NYU, School of Medicine, New York, NY 10016, USA; julia.citron@nyulangone.org (J.C.); Emma.Willcocks@nyulangone.org (E.W.); George.Crowley@nyulangone.org (G.C.); Sophia.Kwon@nyulangone.org (S.K.); 2Department of Environmental Medicine, New York University, School of Medicine, New York, NY 10016, USA; 3Bureau of Health Services, Fire Department of New York, Brooklyn, NY 11201, USA

**Keywords:** genomics, epigenetics, particulate matter, cardiopulmonary disease

## Abstract

Particulate matter (PM) exposure is associated with the development of cardiopulmonary disease. Our group has studied the adverse health effects of World Trade Center particulate matter (WTC-PM) exposure on firefighters. To fully understand the complex interplay between exposure, organism, and resultant disease phenotype, it is vital to analyze the underlying role of genomics in mediating this relationship. A PubMed search was performed focused on environmental exposure, genomics, and cardiopulmonary disease. We included original research published within 10 years, on epigenetic modifications and specific genetic or allelic variants. The initial search resulted in 95 studies. We excluded manuscripts that focused on work-related chemicals, heavy metals and tobacco smoke as primary sources of exposure, as well as reviews, prenatal research, and secondary research studies. Seven full-text articles met pre-determined inclusion criteria, and were reviewed. The effects of air pollution were evaluated in terms of methylation (*n* = 3), oxidative stress (*n* = 2), and genetic variants (*n* = 2). There is evidence to suggest that genomics plays a meditating role in the formation of adverse cardiopulmonary symptoms and diseases that surface after exposure events. Genomic modifications and variations affect the association between environmental exposure and cardiopulmonary disease, but additional research is needed to further define this relationship.

## 1. Introduction

The adverse effects of particulate matter (PM) exposure are a global health concern. PM is a heterogeneous mixture classified by aerodynamic diameter as fine (≤2.5 µm) or coarse (2.5–10 µm) [1]. Respirable PM can cause damage to multiple organs, primarily to the lungs [2,3]. Wood smoke, black carbon, and traffic-related air pollutants (TRAP) have also been associated with increased risk of lung injury [4,5,6,7]. Concern of exposure to PM and other air pollutants was heightened after the events that took place on September 11th, 2001 (9/11). More than 300,000 workers, local residents, and rescue workers were exposed to about 10 million tons of material that was aerosolized after the collapse of the World Trade Center (WTC) [8]. World Trade Center particulate matter (WTC-PM) differed from ambient PM in composition and in its significantly higher concentration; it was comprised of debris from construction buildings containing concrete, pulverized glass, alkaline metals, asbestos, and components of jet fuel [9,10].

Our group has extensively examined the effects of WTC-PM on firefighters. Although we have shown multiple mediators linking exposure and WTC lung injury (WTC-LI; defined as FEV_1_% predicted less than the lower limit of normal), there is a lack of knowledge of how genomics mediates this relationship [8,11,12,13]. Genomics research analyzes the genome, aiming to understand its function and mark changes [14]. This work has provided vital information regarding disease development and continues remains important in the early detection of disease [15,16,17,18]. The association between genetic variations, epigenetic markers, and risk of disease is a developing area of research. Since the genome can be affected by environmental interactions, knowledge of the relationship between the genome and exposure to air pollutants can help to establish future preventative and therapeutic measures [19].

Recent studies have dissected the relationship between the genome and disease state after adverse environmental exposure events [20,21,22]. Within a cohort of subjects present during the WTC attacks, varying expression of genes involved in the hypothalamic-pituitary-adrenal axis was linked to post-traumatic stress disorder [23]. A study of firefighters exposed to smoke and debris identified 17 variants of human leukocyte antigen (HLA) and non-HLA genes associated with sarcoidosis [21]. These studies highlight the importance of genomics as a key component of the systems biology approach of disease understanding. However, there is limited research on the specific association between exposure to air pollution and genomic effects as it relates to cardiopulmonary disease. Cardiopulmonary disease is of concern, due to the rising number of cases in PM-exposed populations [9,24,25,26]. The purpose of this manuscript is to review the literature published over the prior 10 years that investigate the genomics of PM-exposure related cardiopulmonary disease. This review will help advance subsequent studies examining WTC-PM exposure, which includes PM of a heterogeneous aerodynamic diameter. This is in contrast to other more recent reviews on general pollution (including those in soil and water), or fine particle environmental pollution, and cardiovascular disease [27,28].

## 2. Methods 

### 2.1. Eligibility/Study Selection 

PubMed was searched on 6 November 2019, for the Search Terms as per Table 1. Results of this Search (*n* = 95) are in Appendix A. Titles and abstracts (*n* = 65) screened by AN, JC, and EW, Appendix A. 

### 2.2. Definitions

Genomic modification was defined as any genetic or allelic variants, as well as signs of oxidative stress marked at specific loci. Epigenetic modification was defined as the addition or deletion of epigenetic markers that altered the expression of cardiac or respiratory related genes. In this review, cardiopulmonary disease was defined to include obstructive airways disease (OAD), chronic obstructive pulmonary disease (COPD), asthma, chronic bronchitis, emphysema, heart failure, coronary artery disease, congenital heart disease, and peripheral artery disease. Environmental exposure was defined as non-tobacco smoke, PM, and TRAP.

### 2.3. Search Terms

PubMed was queried (06/11/2019) for publications about genomics in relation to cardiopulmonary disease and exposure to air pollutants, utilizing the following search terms:

(particulate matter OR air pollution OR atmospheric pollution OR environmental pollution OR air pollutants OR atmospheric pollutants OR environmental pollutants) AND (genetic modification OR epigenetic modification) AND (lung diseases OR obstructive lung disease OR obstructive airway disease OR obstructive airways disease OR asthma OR chronic bronchitis OR COPD OR chronic obstructive pulmonary disease OR emphysema OR cardiovascular disease OR heart failure OR coronary artery disease OR congestive heart failure OR congenital heart disease OR peripheral artery disease OR metabolic syndrome), Table 1.

### 2.4. Inclusion/Exclusion Criteria

We included original articles that examined: i. lung disease; ii. heart disease; iii. epigenetic modification; iv. specific genes/alleles and; v. PM/air pollution. We excluded studies that were: i. reviews/statements; ii. abstracts; iii. no full text available; iv. secondary research on a previous study; v. tobacco/cigarette smoke as a primary source of exposure; vi. prenatal; vii. occupational exposure; viii. heavy metal exposure; ix. lacking mention of a specific allele or loci and; x. not related to genomics/epigenetic modification. Results from the database search were imported into EndNoteX9 (Clarivate Analytics, Philadelphia, PA, USA).

### 2.5. Data Extraction

In the initial screening, two researchers (Julia Citron (J.C.), Emma Willcocks (E.W.)) screened titles and abstracts for relevant articles. Then, three trained researchers (J.C., E.W., and Anna Nolan (A.N.)) independently reviewed the remaining full-text articles for eligibility. In both screenings, disagreements were resolved through unanimous consensus. Relevant data on genomics, air pollution exposure, and cardiopulmonary disease were compiled to include the specific environmental exposure, cohort characteristics, type of genetic variation or epigenetic modification, and relevant outcomes.

### 2.6. Methods to Limit Bias

Selection bias was limited by defining fixed inclusion/exclusion criteria and by defining intentional subject matter. Detection bias was addressed by having two individuals screen the search results separately. Reporting bias was limited by using PubMed search filters, screening for peer-reviewed published articles written within 10 years.

## 3. Results

### 3.1. Study Selection and Characteristics

The initial PubMed search yielded 95 publications, Figure 1. After screening for studies published within 10 years, 65 papers remained and were assessed for inclusion. During initial title/abstract screening, based on pre-determined criteria, 11 publications were excluded. Exclusion of the remaining full-text articles (*n* = 54) was based on: i. full-text not available *(n* = 4); ii. review/statement (*n* = 23); iii. cigarette/tobacco smoke as a primary source of exposure (*n* = 8); iv. prenatal studies (*n* = 4); v. occupational exposure (*n* = 3); vi. heavy metal exposure (*n* = 2); vii. no specific allele or loci (*n* = 1) and; viii. secondary research (*n* = 2), Figure 1 and Appendix A.

The final analysis incorporates seven full-text publications, Table 2: three focused on deoxyribonucleic acid (DNA) methylation and its effects on lung function and pro-inflammatory markers; two focused on oxidative stress after PM exposure and its association with heart defects; and two studied the effects of pollution in populations with genetic variants.

### 3.2. Effect of Methylation on Lung Function and Inflammation

After air pollution exposure, change in DNA methylation has been correlated with loss of lung function and inflammation [29,32,33]. The work of Sood et al. (2010) used a cohort of smokers with wood smoke exposure to identify that the methylated p16 gene and methylated GATA4 gene were associated with a lower percentage of predicted FEV_1_ [33]. A methylated GATA4 gene was also associated with a greater chance of airflow obstruction, defined by the Global Initiative for Chronic Obstructive Lung Disease (GOLD) criteria as FEV_1_/FVC less than 70% [33]. Wood smoke exposure was an overall predictor of COPD, after adjusting for cigarette smoking, and was therefore independently linked with a higher risk of respiratory disease [33].

Somineni et al. (2016) investigated the role of the Ten-Eleven Translocation 1 (TET1) enzyme that regulates DNA methylation in asthma development [29]. A cohort of African American children was found to be at higher risk of developing asthma, after exposure to TRAP when a 5’-C-phosphate-G-3’(CpG) site in the TET1 promoter was methylated [29]. The association between asthma status and cg23602092 methylation was statistically significant (*p* = 0.040) [29]. This CpG site shows potential as a biomarker for asthma, because cg23602092 methylation was correlated across nasal cells, saliva, and peripheral blood mononuclear cells (PBMC’s) (Pearson’s r ≥ 0.75) [29].

Predisposition to the adverse effects of air pollution exposure was also signified by methylation of LINE-1, Alu, F3, TLR-2, and ICAM-1 in a study of an elderly cohort [32]. Air pollution was linked to changes in markers of coagulation (fibrinogen), inflammation (C-reactive protein), and endothelial function (ICAM-1 and VCAM-1), that may influence risk of cardiovascular disease [32]. There were greater effects of TRAP on fibrinogen and C-reactive protein for those who had either lower methylation of LINE-1 or higher methylation of Alu [32].

### 3.3. Oxidative Stress Score Associated with Cardiac Deficiency

Oxidative stress at specific loci was another marker of distress after PM exposure, and was correlated with cardiovascular defects [30,34]. QT duration, or the duration of time between the Q and T wave on an electrocardiogram, is a parameter that represents the duration of time for ventricular depolarization and repolarization. Prolongation can lead to fatal arrhythmia and sudden cardiac death. The work of Mordukhovich et al. (2016) described stable associations of QT prolongation with increase of PM_2.5_ exposure over time in an elderly cohort [30]. The same group found that long-term PM_2.5_ exposure had the greatest effect on QT, with a 1 year moving average of 9.8µg/m^3^ of PM_2.5_ increasing QT duration by as much as 6.3 ms [30]. They found further associations with genetic variants of *CAT*, *GC*, *GCLM*, *HMOX-1*, and *NQO1* genes related to oxidative damage [30].

The oxidative stress response to PM_2.5_ and black carbon exposure was studied via genetic variants of GSTM1, GSTP1, GSTT1, NQO1, catalase, and HMOX-1 [34]. Black carbon is PM formed through the incomplete combustion of fossil fuels and biomass, and is a major contributor to global climate change. Measurements of systolic blood pressure (SBP) and diastolic blood pressure (DBP) were collected from each participant, to understand the effects of air pollution on cardiac function. Increasing black carbon concentration by 0.43 μg/m^3^ led SBP to increase by 1.46-mmHg (95% CI: 0.10, 2.82) and DBP to increase by 0.87-mmHg (95% CI: 0.15, 1.59). In contrast, PM_2.5_ did not have the same effect on SBP or DBP. However, the authors state that the data were limited, because stationary measurements were used to collect data on PM concentrations, which could underestimate PM concentration, whereas longitudinal measures have been demonstrated to more accurately capture variations in everyday exposure [36,37]. Additionally, the relationship between antioxidant-defense-related genetic variants and modifications to blood pressure (BP) after exposure to black carbon or PM_2.5_ was not statistically significant. However, this is likely attributed to the study’s limitations, as their power to detect a relative effect modification of two was 47% power for a gene prevalence of 20% and 67% power for a gene prevalence of 50% [29].

### 3.4. Genetic Variation Mediates Effect of Air Pollution

The work of Hwang et al. (2013) looked at the effect of PM_2.5_ exposure on participants with different alleles of genes in the glutathione-S-transferase (GST) superfamily [31]. Members of the GST superfamily are of interest because they are expressed in the respiratory tract and are related to asthma pathogenesis [38]. Increased risk of asthma after exposure to PM_2.5_ and O_3_ was found in allelic variants of val105. Specifically, Ile-105 homozygotes displayed a significant, negative association between risk of asthma and PM_2.5_ exposure, as well as between asthma and O_3_ exposure [31]. In addition, Ile-105 homozygotes had a reduced risk of wheezing after PM_2.5_ and O_3_ exposure, compared to those with at least one val105 allele [31].

The relationship between genetic composition and air pollution was further studied in a cohort of myocardial infarction (MI) survivors [35]. Genetic variants of fibrinogen and interleukin-6 (IL-6) were of central focus; genomic analysis incorporated single-nucleotide polymorphisms (SNPs) of IL-6, the fibrinogen α-chain gene (FGA), and the β-chain gene (FGB). The amount of IL-6 found in plasma was recorded, to note the levels of inflammatory response correlated with cardiovascular abnormalities [35]. The most significant change was found 6–11 h after carbon monoxide exposure, in those with the IL-6 rs2069832 variant and the FGB rs1800790 variant [35]. There was also a slight change in plasma IL-6 after exposure to NO_2_, PM_2.5_, and PM_10_, but the change was not as clearly defined [35]. There was also a marked difference between carriers of the minor versus major allele types for these expressed polymorphisms [35]. A significant increase in plasma IL-6 levels was found in participants carrying the minor alleles of IL-6 rs2069832 and IL-6 rs2069845, as well as the major allele of IL-6 rs2069840 (*p* < 0.05) [35].

## 4. Discussion

This review includes seven articles in which genomics has been identified as an important factor for distinguishing the effects of exposure to air pollution. The application of genomics is promising for its ability to differentiate individuals at greater risk for disease [39,40,41]. The concept of genetic vulnerability has been previously explored, with diseases like breast and ovarian cancer, as well as with type 2 diabetes [42,43,44]. These studies have provided information for early disease detection that can be used to improve patient care and impact patient outcomes.

Genomic analysis is relevant in the field of cardiopulmonary research, and the evaluation of pre-determined genetic or allelic variations within a population has been successful in defining at-risk individuals [31,35]. As described in this review, carriers of different alleles in the GST superfamily had significantly different responses to PM_2.5_ exposure, and some carriers were at greater risk for developing asthma [31]. Similar findings were seen in those with genetic variants of fibrinogen and IL-6. Different SNPs of IL-6 and fibrinogen resulted in a greater inflammatory response after exposure to air pollution, specifically carbon monoxide, where inflammatory response marked cardiovascular abnormalities [35]. Our group has completed prior analyses on markers of inflammation, including IL-6. Biomarkers of inflammation have been used to identify cases of WTC-LI in a cohort of never smokers who were exposed to WTC-PM exposure [45]. Information on how the genome impacts these pro-inflammatory biomarkers can strengthen our future studies on WTC-PM exposed individuals and the detection of WTC-LI. The ability to detect subpopulations that are especially vulnerable to adverse cardiopulmonary effects provides greater potential to avoid permanent damage, as it allows for quicker implementation of a treatment plan following exposure.

Risk of disease is also affected by external factors, and symptoms can develop after environmental exposures [46,47]. Gene–environment interaction is known to have serious effects on health, and is applicable when discussing exposure-associated cardiopulmonary disease [48,49,50]. Detection of epigenetic modifications that occur after an exposure event has been a key feature of genomics research [51]. Studies have shown how environmental pollutants can alter the presence of DNA methylation [52,53]. Differences in methylation were associated with asthma, COPD, and inflammation [29,32,33]. Specifically, heightened methylation at distinct loci was correlated with low FEV_1_ function, increased risk of airflow obstruction, and increased risk of childhood asthma [29,33]. A third study reported that decreased methylation at one site was associated with an increase in biomarkers, markers of cardiovascular disease, whereas increased methylation at a separate site was also associated with an increase in these biomarkers [32]. Though there was variation between the amounts of methylation that displayed a significant effect, the study supports the fact that overall level of methylation is a relevant factor in determining exposure susceptibility. DNA methylation regulates expression of genes and is therefore important in gene interactions and subsequent effects on the body [54]. In the studies analyzed, methylation could impact on cardiopulmonary health via this epigenetic mechanism.

Oxidative stress levels are additionally relevant in determining cardiopulmonary health. Reactive oxygen species (ROS) collect within the body and can cause damage to DNA and other cellular structures [55]. Oxidative stress at specific loci was used to identify individuals at risk of disease. Long-term PM_2.5_ exposure was associated with a prolonged corrected QT duration that was further related to an oxidative stress score, calculated using variants of five oxidative defense genes [30]. Greater genetic vulnerability to oxidative stress leaves subjects at risk of inflammation and can interrupt the autonomic cardiac control that regulates QT duration [30]. However, there was no modification effect of oxidative stress on the association between BP and black carbon exposure [34]. Although the study states that no significant relationship was detected, this may be due to its limited power of detection, and, therefore, more oxidative stress genomics studies are needed before it is possible to conclude its effects on cardiac function.

This review has inherent limitations that were minimized by having clearly defined search criteria. Selection bias was addressed by having pre-determined inclusion/exclusion criteria, in addition to having results screened independently by two individuals. Reporting bias was minimized by selecting for peer-reviewed articles published within the past 10 years on PubMed. There is also bias due to the limited data available on the genomics of exposure-associated cardiopulmonary disease, as genomics is a relatively new field.

This review examines only PM exposure. Although air pollution includes gaseous pollutants that have been linked to cardiopulmonary health, such as ozone, PM is a significant component. Further, although WTC-PM is distinct from ambient PM in its composition and concentration levels, we do not distinctly separate them as PM exposure effects on epigenetics. Nevertheless, the limitations of this study do not affect the clear relevance of genomics research. Analyzing the effects of air pollution exposure through the identification of pre-determined genetic variants, epigenetic modifications, and levels of oxidative stress, has clinical importance. Further research will allow for greater generalization and application. Genomics research has the potential for future impact on PM exposure studies.

## 5. Conclusions

Genomics research has the potential for significant impact on exposure-associated cardiopulmonary disease. Both the level of methylation and pre-determined genetic variations were shown to significantly mediate the relationship between exposure events and disease state. The effect of oxidative stress on the relationship between exposure events and disease state, in contrast, was inconclusive, indicating that further research is necessary. Nevertheless, there was a consistent, negative impact of environmental exposure on cardiopulmonary health, particularly with regard to COPD, asthma, BP, QT duration, and inflammation.

## Figures and Tables

**Figure 1 ijerph-16-04335-f001:**
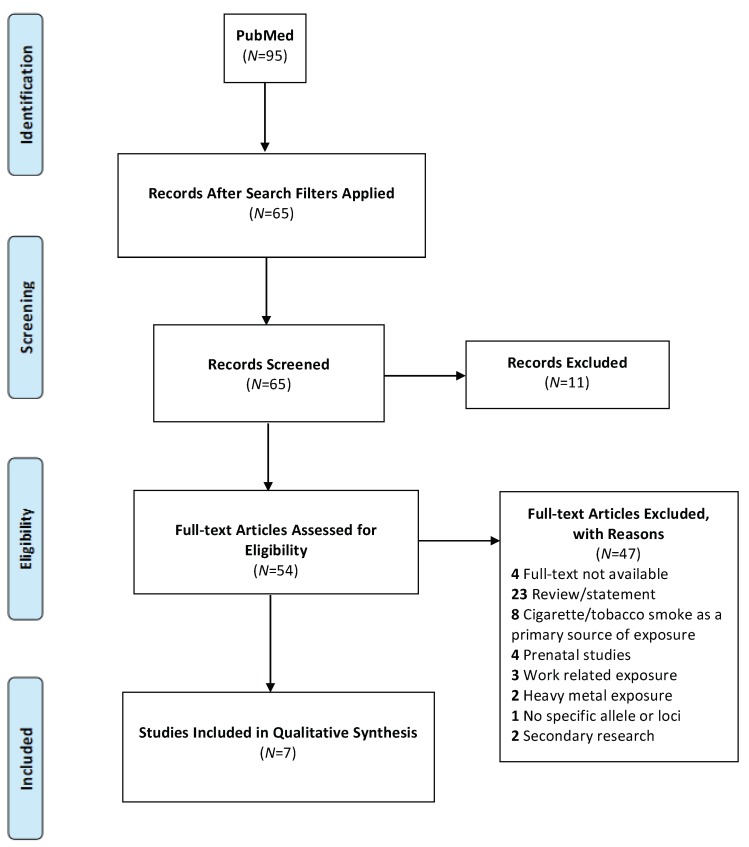
Study Design and Article Extraction.

**Table 1 ijerph-16-04335-t001:** Search Terms.

**PubMed Search**	(particulate matter OR air pollution OR atmospheric pollution OR environmental pollution OR air pollutants OR atmospheric pollutants OR environmental pollutants) AND (genetic modification OR epigenetic modification) AND (lung diseases OR obstructive lung disease OR obstructive airway disease OR obstructive airways disease OR asthma OR chronic bronchitis OR COPD OR chronic obstructive pulmonary disease OR emphysema OR cardiovascular disease OR heart failure OR coronary artery disease OR congestive heart failure OR congenital heart disease OR peripheral artery disease OR metabolic syndrome)

Abbreviation: COPD: chronic obstructive pulmonary disease.

**Table 2 ijerph-16-04335-t002:** Study Characteristics.

First Author [ref]	Year	Environmental Exposure	Outcome Measure	Result
**Somineni** [29]	2016	TRAP	Asthma	cg23606092 methylation was associated with increased risk of childhood asthma
**Mordukhovich** [30]	2016	PM_2.5_	QT duration	High allelic risk profiles calculated based on the genetic variants of CAT, GC, GCLM, HMOX-1, and NQO1 were associated with increased QT duration
**Hwang** [31]	2013	Sulfur dioxide, nitrogen dioxides, ozone, carbon monoxide and PM_2.5_	Asthma/wheezing	Ile105 carriers were positively associated with risk of asthma after PM_2.5_ and O_3_ exposure
**Bind** [32]	2012	Black carbon, carbon monoxide, sulfate, nitrogen dioxide, ozone and PM_2.5_	Inflammatory biomarkers	Lower methylation of LINE-1 and higher methylation of Alu were both associated with biomarkers of cardiovascular disease
**Sood** [33]	2010	Wood smoke	COPD	Methylation at the promoter region of p16 predicted lower FEV_1_ function and methylation at the promoter region of GATA4 was associated with airflow obstruction; wood smoke was an overall predictor of COPD
**Mordukhovich** [34]	2009	Black carbon and PM_2.5_	Blood pressure	Black carbon was associated with increased BP and there was no association with antioxidant-defense-related genetic variants
**Ljungman** [35]	2009	Carbon monoxide, nitrogen dioxide, PM_10_ and PM_2.5_	Inflammatory biomarkers	Air pollution has a greater effect on those with SNPs IL-6 rs2069832 and FGB rs1800790

Abbreviations: BP: Blood Pressure; COPD: Chronic Obstructive Pulmonary Disease; FEV_1_: Forced Expiratory Volume Over One Second; FGB: Fibrinogen β -Chain Gene; HMOX-1: Heme Oxygenase-1; IL-6: Interleukin-6; NQO1: NAD(P)H Quinine Oxidoreductase 1; PM: Particulate Matter; SNP: Single-Nucleotide Polymorphism; TRAP: Traffic-Related Air Pollution.

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
