# Peer review of "Genomics of Particulate Matter Exposure Associated Cardiopulmonary Disease: A Narrative Review"

_ijerph, 2019, doi:10.3390/ijerph16224335_

Round 1

Reviewer 1 Report

This is an important area that needs more research, and understanding the published material is important to encourage further studies. The authors have done a timely review, however, some major and minor considerations need to be addressed.

Major Comments:

I believe the authors should establish why this review is important, compared to other reviews that have shown air pollution exposure and genomic changes. There are recent reviews published that have shown similar associations and discussed the importance. The authors should mention why this review will add more to the our understanding (or the lack of) about the subject. The topic mentions 'Genomics of particulate matter exposure..', however, most the studies highlighted, content in the discussion, and conclusion are mainly about overall air pollution exposure (PM and gaseous pollutants and/or a mixture of them) . Most studies have gaseous air pollutant exposures, which are also associated with cardiopulmonary health (particularly Ozone). If they main body of work was on both gaseous and particulate matter pollutants, then I believe the topic should be changed accordingly, or discuss this as a limitation - particularly the impact of Ozone and other gases on cardiopulmonary health and andgenetic changes 

Minor comments:

Line 49: Support this statement by providing reference/s I believe it is important to mention that the exposures experienced in the WTC disaster is multiple times higher than normal ambient PM exposures and the impacts of the current studies only reflect outcomes due to exposure to a certain range of concentrations Discuss confounding by other air pollutants if PM is the focus (refer to major comment 2 above) Line 128: The method mentions tobacco smoke exposures were excluded. This study mentions female smokers Lines 153 and 159: Give units for PM (A 4.0 increase.. of what units?) Suggest to define QT duration at the beginning Line 182: Not very important, but I would prefer if instead of saying 'One study..' to say '[Author name] et al (year) looked at the effect... because it is only one study

Author Response

REVIEWER 1
This is an important area that needs more research, and understanding the published material is important to encourage further studies. The authors have done a timely review, however, some major and minor considerations need to be addressed.

Major Comments:
I believe the authors should establish why this review is important, compared to other reviews that have shown air pollution exposure and genomic changes. There are recent reviews published that have shown similar associations and discussed the importance. The authors should mention why this review will add more to our understanding (or the lack of) about the subject.

We have included in the introduction a statement to clarify how our review differs from other recently published reviews (lines 60-65). Furthermore we have include new references specific to these:

“The purpose of this manuscript is to review literature published over the prior 10 years that investigate the genomics of PM-exposure related cardiopulmonary disease. This review will help advance subsequent studies examining WTC-PM exposure, which included PM of a heterogeneous aerodynamic diameter. This is in contrast to other more recent reviews on general pollution (including those in soil and water) or fine particle environmental pollution and cardiovascular disease.1,2

The topic mentions 'Genomics of particulate matter exposure.', however, most the studies highlighted, content in the discussion, and conclusion are mainly about overall air pollution exposure (PM and gaseous pollutants and/or a mixture of them). Most studies have gaseous air pollutant exposures, which are also associated with cardiopulmonary health (particularly Ozone). If the main body of work was on both gaseous and particulate matter pollutants, then I believe the topic should be changed accordingly, or discuss this as
a limitation - particularly the impact of Ozone and other gases on cardiopulmonary health and genetic changes.

We now include the following to address these limitations in line 250-254.

“This review examines only PM exposure. Although air pollution includes gaseous pollutants that have been linked to cardiopulmonary health such as ozone, PM is a significant component. Further, although WTC-PM is distinct from ambient PM in composition and in concentration levels, we do not distinctly separate them as PM exposure effects on epigenetics. “

Minor comments:
Line 49: Support this statement by providing references. I believe it is important to mention that the exposures experienced in the WTC disaster is multiple times higher than normal ambient PM exposures and the impacts of the current studies only reflect outcomes due to exposure to a certain range of concentrations

We have examined the following statement as the one in question on Line 46: Association between congenital genetic variations and risk of disease development, as well as the association between epigenetic markers and risk of disease state is promising, but this work remains to be validated.

We have changed this sentence for clarity, and to reflect that epigenetics is a developing field. We now state “Association between genetic variations, epigenetic markers, and risk of disease is a developing area of research. Since the genome can be affected by environmental interactions, knowledge of the relationship between the genome and exposure to air pollutants can help establish future preventative and therapeutic measures.3

We also cite Garcia-Gimenez et al‘s paper on the current state of epigenetics as a reference.

We also include the following statement to highlight the unique WTC exposure disaster:

Line 38: WTC-PM differed from ambient PM in composition and in the significantly higher concentration; it was comprised of debris from construction buildings containing concrete, pulverized glass, alkaline metals, asbestos, and components of jet fuel 4,5.

And:

The purpose of this manuscript is to review literature published over the prior 10 years that investigate the genomics of PM-exposure related cardiopulmonary disease. This review will help advance subsequent studies examining WTC-PM exposure, which included PM of a heterogeneous aerodynamic diameter. This is in contrast to other more recent reviews on general pollution (including those in soil and water) or fine particle environmental pollution and cardiovascular disease.1,2

Discuss confounding by other air pollutants if PM is the focus (refer to major comment 2 above)

We included a more detailed limitation section starting in line 250.

“This review examines only PM exposure. Although air pollution includes gaseous pollutants that have been linked to cardiopulmonary health such as ozone, PM is a significant component. Further, although WTC-PM is distinct from ambient PM in composition and in concentration levels, we do not distinctly separate them as PM exposure effects on epigenetics. “

Line 128: The method mentions tobacco smoke exposures were excluded. This study mentions female smokers Lines 153 and 159:

We have clarified in our exclusion criteria. We now state that we excluded studies investigating smoking as a primary source of exposure. The study in question, Sood et al (2010), focused on wood smoke exposure in a population of cigarette smokers. We have edited line 93 from the inclusion/exclusion criteria to read: v. tobacco/cigarette smoke as a primary source of exposure

Give units for PM (A 4.0 increase. of what units?)

Thank you for bringing this paragraph to our attention. We found that in order to improve clarity, the sentence structure benefited from some pruning of extraneous numbers. We have adjusted the statement to say the following at line 154:

Mordukhovich et al (2016) described stable associations of QT prolongation associated with increase of PM2.5 exposure over time in an elderly cohort 6. The same group found that long-term PM2.5 exposure had the greatest effect on QT, with a 1 year moving average of 9.8μg/m3 of PM2.5 increasing QT duration by as much as 6.3ms6. Further, they found associations with genetic variants of CAT, GC, GCLM, HMOX-1, and NQO1, genes related to oxidative damage 6.

Suggest to define QT duration at the beginning

Thank you, we have clarified the definition as the following at line 151:

QT duration is an electrocardiogram parameter representing the duration of time for ventricular depolarization and repolarization that prolongation can lead to fatal arrhythmia and sudden cardiac death.

Line 182: Not very important, but I would prefer if instead of saying 'One study.' to say '[Author name] et al (year) looked at the effect... because it is only one study

We have taken your suggestion and changed the sentence to read in line 177:

Hwang et al (2013) looked at the effect of PM2.5 exposure on participants with different alleles of genes in the glutathione-S-transferase (GST) superfamily 7.

And also in line 154: Mordukhovich et al (2016) described stable associations of QT prolongation associated with increase of PM2.5 exposure over time in an elderly cohort.

We also changed line 136 to include only one reference as the sentence was discussing only one aspect of TET enzymes: Somineni et al (2016) investigated the role of the Ten-Eleven Translocation 1 (TET1) enzyme that regulates DNA methylation in asthma development 8

REFERENCES

1. Combes A, Franchineau G. Fine particle environmental pollution and cardiovascular diseases. Metabolism. 2019;100S:153944.
2. Hu JJ, Yu YX. Epigenetic response profiles into environmental epigenotoxicant screening and health risk assessment: A critical review. Chemosphere. 2019;226:259-272.
3. Garcia-Gimenez JL, Seco-Cervera M, Tollefsbol TO, et al. Epigenetic biomarkers: Current strategies and future challenges for their use in the clinical laboratory. Crit Rev Clin Lab Sci.  2017;54(7-8):529-550.
4. Lippmann M, Cohen MD, Chen LC. Health effects of World Trade Center (WTC) Dust: An unprecedented disaster's inadequate risk management. Crit Rev Toxicol. 2015;45(6):492-530.
5. Landrigan PJ. Health consequences of the 11 September 2001 attacks. Environ Health Perspect. 2001;109(11):A514-515.
6. Mordukhovich I, Kloog I, Coull B, Koutrakis P, Vokonas P, Schwartz J. Association Between Particulate Air Pollution and QT Interval Duration in an Elderly Cohort. Epidemiology (Cambridge, Mass). 2016;27(2):284-290.
7. Hwang BF, Young LH, Tsai CH, et al. Fine particle, ozone exposure, and asthma/wheezing: effect modification by glutathione S-transferase P1 polymorphisms. PLoS One. 2013;8(1):e52715.
8. Somineni HK, Zhang X, Biagini Myers JM, et al. Ten-eleven translocation 1 (TET1) methylation is associated with childhood asthma and traffic-related air pollution. J Allergy Clin Immunol. 2016;137(3):797-805 e795.
9. Rojas-Bracho L, Suh HH, Koutrakis P. Relationships among personal, indoor, and outdoor fine and coarse particle concentrations for individuals with COPD. J Expo Anal Environ Epidemiol. 2000;10(3):294-306.
10. Nethery E, Teschke K, Brauer M. Predicting personal exposure of pregnant women to traffic-related air pollutants. Sci Total Environ. 2008;395(1):11-22.

Reviewer 2 Report

This is a well-written and good piece of work. 

My comments are rather organizational and style related.

The tables need some organization so that the width of the column contains a full word and the word is not splitted into tow rows.

The abbreviations: I think that there is no need to list the abbreviations at the end of the manuscript. You have already included them in each part they are mentioned and in the footer of the tables. Just make sure that each word is spelled out completely in the first mention with the abbreviation in brackets. And then in the second mention you could just use the abbreviation. In this case, there is no need to list them again in the paper.

Best wishes

Author Response

REVIEWER 2

This is a well-written and good piece of work. My comments are rather organizational and style related. The tables need some organization so that the width of the column contains a full word and the word is not splitted into two rows.

Thank you we have taken your suggestion and have made an adjustment to Table 2 to accommodate the
width.

We have also made the following adjustments to the text for clarity in the following lines:

Line 38: WTC-PM differed from ambient PM in  composition and in the significantly higher  concentration; it was comprised of debris from construction buildings containing concrete, pulverized glass, alkaline metals, asbestos, and components of jet fuel 4,5.

Line 95: v. tobacco/cigarette smoke as a primary source of exposure;

Line 165: In contrast, PM2.5 did not have the same effect on SBP or DBP. However, the authors state that the data was limited because stationary  measurements were used to collect data on PM concentrations which could underestimate PM concentration, whereas longitudinal measures have been demonstrated to more accurately capture variations in everyday exposure 9,10.

The abbreviations: I think that there is no need to list the abbreviations at the end of the manuscript. You have already included them in each part they are mentioned and in the footer of the tables. Just make sure that each word is spelled out completely in the first mention with the abbreviation in brackets.

Thank you for your suggestion. We have removed the abbreviation list.

REFERENCES

1. Combes A, Franchineau G. Fine particle environmental pollution and cardiovascular diseases. Metabolism. 2019;100S:153944.
2. Hu JJ, Yu YX. Epigenetic response profiles into environmental epigenotoxicant screening and health risk assessment: A critical review. Chemosphere. 2019;226:259-272.
3. Garcia-Gimenez JL, Seco-Cervera M, Tollefsbol TO, et al. Epigenetic biomarkers: Current strategies and future challenges for their use in the clinical laboratory. Crit Rev Clin Lab Sci.  2017;54(7-8):529-550.
4. Lippmann M, Cohen MD, Chen LC. Health effects of World Trade Center (WTC) Dust: An unprecedented disaster's inadequate risk management. Crit Rev Toxicol. 2015;45(6):492-530.
5. Landrigan PJ. Health consequences of the 11 September 2001 attacks. Environ Health Perspect. 2001;109(11):A514-515.
6. Mordukhovich I, Kloog I, Coull B, Koutrakis P, Vokonas P, Schwartz J. Association Between Particulate Air Pollution and QT Interval Duration in an Elderly Cohort. Epidemiology (Cambridge, Mass). 2016;27(2):284-290.
7. Hwang BF, Young LH, Tsai CH, et al. Fine particle, ozone exposure, and asthma/wheezing: effect modification by glutathione S-transferase P1 polymorphisms. PLoS One. 2013;8(1):e52715.
8. Somineni HK, Zhang X, Biagini Myers JM, et al. Ten-eleven translocation 1 (TET1) methylation is associated with childhood asthma and traffic-related air pollution. J Allergy Clin Immunol. 2016;137(3):797-805 e795.
9. Rojas-Bracho L, Suh HH, Koutrakis P. Relationships among personal, indoor, and outdoor fine and coarse particle concentrations for individuals with COPD. J Expo Anal Environ Epidemiol. 2000;10(3):294-306.
10. Nethery E, Teschke K, Brauer M. Predicting personal exposure of pregnant women to traffic-related air pollutants. Sci Total Environ. 2008;395(1):11-22.